## [Peer Review File · Nature Genetics]

GGC repeat expansions within novel open reading frames are translated into toxic polyglycine proteins in oculopharyngodistal myopathy

Corresponding Author: Dr Nicolas Charlet-Berguerand

This manuscript has been previously reviewed at another journal. This document only contains information relating to versions considered at Nature Genetics.

Version 0:

Decision Letter:

31st May 2025

Dear Nicolas,

Your Article "Microsatellite expansions hidden within the human dark genome are translated in novel and toxic proteins causing muscle and neurodegenerative diseases" has been seen by two referees. You will see from their comments below that, while they find your work of considerable interest, they have raised several relevant points, mostly related to aspects of the presentation. We are interested in the possibility of publishing your study in Nature Genetics, but we would like to see your response to these points in the form of a revised manuscript before we make a final decision on publication.

To guide the scope of the revisions, the editors discuss the referee reports in detail within the team, including with the chief editor, with a view to identifying key priorities that should be addressed in revision, and sometimes overruling referee requests that are deemed beyond the scope of the current study. In this case, we ask that address technical queries related to the constructs used for each experiment, revise the presentation for precision and clarity throughout, and incorporate additional points of discussion as requested by the referees. We hope that you will find this prioritized set of referee points to be useful when revising your study. Please do not hesitate to get in touch if you would like to discuss these issues further.

We therefore invite you to revise your manuscript taking into account all reviewer and editor comments. Please highlight all changes in the manuscript text file. At this stage, we will need you to upload a copy of the manuscript in MS Word .docx or similar editable format.

*2) If you have not done so already, please begin to revise your manuscript so that it conforms to our Article format instructions, available

[here](http://www.nature.com/ng/authors/article_types/index.html).

*3) Include a revised version of any required Reporting Summary: <https://www.nature.com/documents/nr-reporting-summary.pdf>

Please be aware of our [guidelines](https://www.nature.com/nature-research/editorial-policies/image-integrity) on digital image standards.

EXTENDED DATA FIGURES

Link Redacted

We hope to receive your revised manuscript within 4-8 weeks. If you cannot send it within this time, please let us know.

Nature Genetics is committed to improving transparency in authorship. As part of our efforts in this direction, we are now requesting that all authors identified as 'corresponding author' on published papers create and link their Open Researcher and Contributor Identifier (ORCID) with their account on the Manuscript Tracking System (MTS), prior to acceptance. ORCID helps the scientific community achieve unambiguous attribution of all scholarly contributions. You can create and link your ORCID from the home page of the MTS by clicking on 'Modify my Springer Nature account'. For more information, please visit www.springernature.com/orcid.

Sincerely,
Kyle

Kyle Vogan, PhD
Senior Editor
Nature Genetics
<https://orcid.org/0000-0001-9565-9665>

Referee expertise:

Referee #1: Genetics, repeat expansions, neurological disorders

Referee #2: Genetics, repeat expansions, muscle disorders

Reviewers' Comments:

Reviewer #1 (Remarks to the Author):

This manuscript by Manon Boivin and colleagues provides compelling new insights into the pathophysiological mechanisms underlying pathogenic GGC repeat expansions associated with oculopharyngodistal myopathy (OPDM) and oculopharyngeal myopathy with leukodystrophy (OPML).

GGC repeat expansions ranging from approximately 50 to 200 repeats in at least six genes (LOC642361, LRP12, GIPC1, NOTCH2NLC, RILPL1, and ABCD3) have been linked to autosomal dominant disorders spanning a clinical continuum between neuromuscular and neurodegenerative diseases. While the expression of polyGlycine peptides by repeat-associated non-AUG (RAN) translation had previously been demonstrated in Fragile X-associated tremor/ataxia syndrome (FXTAS) and NOTCH2NLC-related neuronal intranuclear inclusion disease (NIID) by the same authors (Boivin et al., Neuron, 2021) and others in the field, this study significantly extends these findings, as follows:

- Mechanism of polyG expression: The authors demonstrate that GGC expansions can lead to polyGlycine peptides through the formation of open reading frames (ORFs) initiated either from canonical ATG start codons or from alternative near-cognate codons. These results challenge the current view of how RAN translation works, which was thought to be

independently of AUG initiation, and shows that translation occurs in selective reading frames, rather than across all three possible frames.

- Distinct polyG proteins were detected in patient tissues using antibodies targeting the unique N-terminal regions of these peptides. These proteins localize to p62-positive intranuclear inclusions, which constitute pathological hallmarks of GGC repeat expansion disorders.

- Ectopic expression of polyG polypeptides in both *Drosophila* and mouse models recapitulates key skeletal muscle pathologies observed in patients, including the formation of p62-positive inclusions.

- Therapeutic potential: importantly, the authors identify a pharmacological compound capable of selectively reducing polyG peptide expression, offering a possible therapeutic strategy for GGC expansion-related diseases.

The manuscript is generally well written, with a clear presentation of a sound experimental approach and robust results.

Suggested improvements: I listed below points that the authors could address to enhance readability and clarify certain aspects of the experimental design and results.

1) The title does not accurately reflect the content of the manuscript and may even be somewhat misleading. It suggests that translation into toxic proteins is a general mechanism associated with microsatellite expansions across the “dark” genome. As the authors correctly acknowledge, >60 disorders associated with repeat expansion have been described, with diverse underlying mechanisms. RAN translation—or translation of repeats into toxic peptides—is only one of them, and many of these disorders are instead caused by gene loss-of-function due to the repeat expansion (e.g., Fragile X, EPM1/CSTB, Friedreich ataxia/FXN, SCA27B/FGF14, GLS, XYLT1, AFF2, AFF3). Furthermore, this study provides no generalizable evidence that the mechanism of translation applies to microsatellite expansions beyond the specific GGC repeats examined. I therefore strongly recommend revising the title to more accurately reflect the results of the study. I also suggest avoiding the use of the terms “dark genome” or “dark proteome”, which also overstates the broader relevance of the findings.

2) Lines 35 and 60-61 (Summary): the first sentence of the summary is potentially misleading, as it suggests that microsatellites make up half of the genome. As correctly stated in the introduction, microsatellites constitute only about 3-6% of the genome and the majority of repetitive DNA consists of other elements such as LINEs, SINEs, and alpha-satellite sequences. The authors should clarify that the mechanisms described in the manuscript are specific to a subset of microsatellite repeat expansions and are not generalizable to other repetitive elements.

3) Lines 104–105: The terms oculopharyngodistal myopathy (OPDM) and oculopharyngeal myopathy with leukodystrophy (OPML) should be spelled out in full and separately at least once in the Introduction, together with their respective OMIM entries. This information currently appears in the Discussion (lines 602–603), but would be more appropriately placed earlier in the manuscript.

4) Lines 154-156 (Introduction): the references cited in the Discussion (lines 653–654) regarding near-cognate codons should also appear in the Introduction to support the earlier mention of this concept.

5) Lines 185-196 (Results): this section of the Results repeats information already presented in the Introduction and could be significantly condensed or combined with other paragraphs elsewhere to avoid redundancy and improve the flow of the manuscript.

6) A recurring issue throughout the manuscript is the use of clinical terms such as “OPDM” and “OPML” to refer to specific GGC repeat expansion constructs rather than specifying the number of repeats and underlying gene. Both OPDM and OPML are phenotypes resulting from GGC repeat expansions in different genes. Similar GGC expansions at the same locus can lead to distinct disease phenotypes, (e.g., NIID or OPDM linked to expansions in NOTCH2NL). To avoid confusion, I strongly recommend that the authors refer to the gene context and repeat numbers when reporting experiments performed with plasmids. “OPDM and OPML GGC repeats” used repeatedly (lines 184, 211, 219, 230, 234, 264...) should be avoided. In Figures 1, 2, and 3, terms like “OPML,” “OPDM2,” and “OPDM4” should be replaced with the specific names of the constructs used, similar to “LOC6polyG”, “uGIPpolyG”, and “asRILpolyG” used in Fig. 2F or Fig. 3A-D. The terms OPDM and OPML could be used when patient-derived material from individuals diagnosed with these specific conditions is used.

7) It is unclear whether the constructs used in Fig. 1C-G, Fig. 2B-D, Fig. 3-7 are the same or if (and how) they differ as the names “LOC6polyG”, “uGIPpolyG”, and “asRILpolyG” are only used from Fig. 2F on. A schematic summary of all the constructs used in the study, including their names, their features (e.g., gene context, number of GGC repeats) and whether they contain an ATG or near-cognate start codons naturally occurring in the human genome, would greatly improve clarity. Ideally, Fig. 1A, 1B, 2A and 2F could be combined in a single figure or revised as needed to clarify how the constructs differ from one another and from those used in previous experiments.

8) Consider revising Fig. 1C-F to incorporate some of the immunofluorescence data and quantifications for GIPC1 and RILPL1 currently shown in Fig. S1. This would ensure consistency with Fig. 1A and 1G, which present data for all three constructs, whereas Panels B-F currently focus only on LOC642361. If space is a limitation, some data (e.g. Panel D) could be moved to the Extended Data.

9) Results, line 21-212: Is there a specific reason why only polyG (and not polyA or polyR) peptides are produced, aside

from reading frame selection and initiation codon context? A brief discussion would be helpful.

10) Text in Fig. 2B-D is not readable.

11) It is currently difficult to clearly distinguish between experiments conducted using in vitro overexpression of GGC repeats from plasmids and those based on patient-derived material. This distinction is critical, as one of the most important contributions of the study is the detection of polyG proteins in patient tissues, which validates that the observed effects are not merely artifacts of constructs/in vitro experiments. The authors should explicitly state which figures correspond to each experimental context by indicating it in the legend of the figure and in the subtitles included in the results. For example, while lines 301–321 suggest that Fig. 3 presents data from patient muscle tissue, the use of the same construct names (e.g., “OPDM2 (uGIPpolyG)” and “OPDM4 (asRILpolyG)” ...) as those used in previous in vitro experiments and the lack of clarity in the figure legend create confusion.

12) Results, lines 342–347: The authors introduce modified constructs containing GGN interruptions, which are intended to stabilize the GGC repeat tracts and reduce their instability. These interruptions likely also alter the RNA secondary structure, potentially impacting RNA toxicity. However, it is unclear in which experiments these interrupted constructs were used. The Results text (lines 342–350) implies their use begins in Figure 4, but since the figure labels and construct names do not specify whether interruptions are present, it remains ambiguous whether comparisons were made with their non-interrupted counterparts.

This is a critical issue, as repeat instability in specific brain tissues has recently been implicated in the pathology of several disorders (e.g., CAG expansions in HTT, PMID: 39824182; AAG expansions in FGF14, PMID: 39378335). Moreover, RNA secondary structures are known to contribute to the disease mechanisms in various repeat expansion disorders such as DM1 and CANVAS. Therefore, it is essential that the authors clarify the use of interrupted constructs and, ideally, compare their impact to that of pure repeats. Alternatively, the authors should provide supporting evidence from patient material demonstrating that GGN-interrupted expansions in OPDM/OPML are pathogenic and associated with comparable clinical phenotypes to those caused by uninterrupted repeats.

13) A recurrent and still unresolved question in the field of repeat expansion disorders associated with nuclear inclusions is whether these inclusions contribute directly to toxicity, or whether they represent a secondary, possibly protective, cellular response—such as sequestration of misfolded or soluble toxic polypeptides. The authors’ findings that polyG proteins exhibit distinct properties, interact with different protein partners, and have varying impacts when delivered via rAAV vectors in animal models could be interpreted as supporting the latter hypothesis. It would be valuable if the authors could comment on this point in the Discussion, particularly in light of their in vivo results.

14) One of the most exciting aspects of this manuscript is the identification of a compound, TMPyP4, that reduces the aggregation and toxicity of polyG proteins. TMPyP4 is best known as a stabilizer of DNA G-quadruplex (G4) structures. Could the authors clarify whether the engineered interruptions introduced into the GGN repeats involve the insertion of adenines (A), which could potentially promote the formation of G4 structures not naturally present in the uninterrupted GGC repeat expansions found in patients? This point is important to assess the relevance of the TMPyP4 findings to the native pathogenic repeats.

15) Discussion, lines 606-607: there any evidence that GGC repeats within the normal range are translated, or do these GGC repeats form an ORF only in pathological/expanded conditions?

16) As briefly discussed in the discussion, not all GGC expansions result in translation into polyG peptides. Only mid-size expansions (50–200 repeats) support translation, while longer expansions are often methylated, leading instead to gene silencing and loss-of-function. It would be helpful if the authors could clearly indicate this distinction and briefly discuss the relationship between expansion size and its functional consequences i.e., translation and hypermethylation-associated silencing a part of the discussion paragraph appearing lines 705-713.

17) Could the authors provide more detail on the expression profiles of the six genes (LOC642361, LRP12, GIPC1, NOTCH2NLC, RILPL1, and ABCD3) in which GGC repeat expansions are linked to the OPDM phenotype? Are all of these genes similarly expressed in both muscle and brain tissues? Do the authors expect the ORFs generated by GGC expansions to follow the expression patterns of their host genes, or could they exhibit distinct regulation? Based on the data presented in this manuscript, could the authors comment on whether differential tissue expression of polyG peptides might underlie phenotypic variability observed across disorders such as OPDM, OPML, NIID, or SCA4 (in the case of GGC expansions in ZFH3).

Reviewer #2 (Remarks to the Author):

This manuscript addresses the fascinating and intriguing genetics of oculopharyngodistal myopathy (OPDM). For some time now it has been known that GGC/CCG repeat expansions in the 5’UTR of six, perhaps seven, different genes cause phenotypically very similar oculopharyngodistal myopathies (OPDM) with or without leukodystrophy (OPML). This has always suggested that the pathomechanism has to do with the GGC/CCG repeat expansions and the genes that the expansions are in the 5’UTR of are, in fact, irrelevant. This manuscript demonstrates that the pathomechanism is transcription and translation of polyglycine-containing proteins from short previously unidentified open reading frames in the 5’UTRs.

The authors have done an extraordinary amount of work.

They focussed their experimental work primarily on three of the OPDM genes: GIPC1, RILPL1 and LOC642361.

They expressed the GGC repeat expansions of GIPC1, RILPL1 and LOC642361 and control size repeats in their genomic context fused to GFP in HEK293 cells in their Glycine, Alanine and Arginine frames. Only the polyglycine frame resulted in protein detectable with GFP antibodies despite RT-qPCR demonstrating similar RNA levels.

They used mass spectrometry to sequence the polyglycine proteins and showed the N-terminal sequences are different, reflecting the sequence of the gene, but all start from a standard acetylated methionine.

Using antibodies to the specific N or C-terminals they demonstrated the presence of the polyglycine proteins for GIPC1, RILPL1 and LOC642361 but also OPDM3 NOTCH2NLC in the p62 inclusions in muscle biopsies from the respective patients.

To test for RAN translation in other potential frames, they developed an antibody specific to the GIPC1 alanine frame but did not see any staining in OPDM2 biopsies.

They expressed the polyglycine proteins in human LHCN-M2 differentiated muscle cells and showed they formed p62 positive cytoplasmic and intranuclear inclusions. There were some differences in the results in that some of the constructs resulted in more cytoplasmic or nuclear inclusions than others.

Immunoprecipitation and mass spectrometry identified interactants that varied between the genes, suggesting the unique N- or C-terminals were involved in the interactions.

Testing the toxicity of the polyglycine proteins in LHCN-M2 differentiated muscle cells showed all were toxic but the RILPL1, NOTCH2NLC and LOC642361 more than GIPC1.

To test pathogenicity in animal models, they expressed the polyglycine proteins in C57BL/6 mice using a myotropic virus MyoAAV 4A which can be used to reach muscle after intravenous injection. This resulted in pathology in both tibialis anterior and gastrocnemius including p62 inclusions. Again, there was variability in cytoplasmic vs intranuclear location between the polyglycine proteins. snRNA sequencing did not show major changes. The asRILpolyG (OPDM4) and LOC6polyG (OPML) polyglycine proteins resulted in early death which was shown to be associated with remarkable dilated cardiomyopathy not shown with the other proteins.

Similarly, to test toxicity in the brain, the authors used an AAV vector PHP.eB rAAV serotype that crosses the mouse blood-brain barrier and targets neurons. This resulted in variably altered performance, loss of neurons including Purkinje cells and early death with the different polyglycine proteins.

In relation to candidate therapies, the authors tested a number of compounds that have been shown to prevent nuclear export of translation of toxic proteins or promote autophagy of aggregation prone proteins. The authors showed the porphyrin TMPyP4 to be a potential therapeutic option for these polyglycine diseases since TMPyP4 reduced toxicity in tissue

Finally in animal models the authors expressed the polyglycine proteins in Drosophila. This resulted in eye, locomotor phenotypes and premature death.

Major comments

To my mind, the major thing missing from the manuscript is a detailed discussion of the pathogenetic mechanism of the diseases. I was expecting in the Discussion a detailed description of what the manuscript demonstrates are the prerequisites for the oculopharyngodistal myopathy phenotype and this detailed explanation is simply not there. The authors demonstrate that the prerequisites are:

There has to be a transcript

There has to be a cognate or near-cognate start codon with an associated Kozak sequence

The start codon has to be in the frame with the GGC repeat that can then be translated into polyglycine.

The authors beautifully demonstrate this in that for the GIPC1 and LOC642361 the (GGC)_n repeat, is in the sense strand, whereas for RILPL1 the sense strand repeat is (CCG)_n and is not translated into polyproline, polyalanine or polyarginine, but there is bidirectional transcription of RILPL1 and it is the antisense transcript encoding a (GGC)_n repeat that gives rise to the polyglycine pathogenic protein.

Their needs to be discussion of why this pathogenetic mechanism occurs in 5'UTR but to date has not been identified in 3'UTR.

In the Discussion, at present there is a very brief discussion (Lines 638-670) that, now that many small ORFs and microsatellites have been identified in the genome, some might be similarly pathogenic to those causing OPMD/OPML. This might be written more succinctly and needs to be offset by the discussion of the stringent multiple requirements for toxic proteins to be produced.

There is no discussion of the fact that multiple other codons in the sense strand could make polyglycine if expanded into a repeat: ie. CCG (Arg), GAG (Glu), GCG (Ala), GGA GGG and GGT (Gly), GTG (Val), or that there are multiple codons if expressed from the reverse strand and expanded into a repeat, could be translated into polyglycine. There should be some discussion as to why this has not been seen.

Medium comments

There should be greater discussion of what the mechanism might be that, despite RT-PCR showing equal expression of the expression vectors, only the polyglycine frame resulted in protein being detectable. What is happening to the polyalanine and polyarginine proteins?

Minor comments

Line 138: Nobody uses "similitudes", similarities would be plainer English.

Line 213: change spelling to "negligible"

Line 213: For Figure 1C and all other figures involving microscopy, please give the magnifications used.

Lines 235-237: "To uncover how these GGC repeats are translated, we immunoprecipitated the OPDM and OPML polyglycine proteins and determined their N-terminal sequences by mass spectrometry analysis."

Please state here in the text that it is from the expression in HEK cells to make it clear to the reader, so the reader doesn't have to search in the Figure legends to find this out.

Line 558: remove 'both'.

Lines 558-559: 'TMPyP4, which efficiently prevents expression of both uGIPpolyG, uN2CpolyG, asRILpolyG and LOC6polyG at the protein level'

Does TMPyP4 prevent expression, or translation? Would it not be better to write that TMPyP4 reduces abundance of the polyG peptides?

This would also be more consistent with the statement later in the paragraph that TMPyP4 acts principally on translation.

Line 606: ABCD3 needs to be added as in the Introduction.

Lines 629-633: 'Short (~30) stretches of polyalanine in various transcription factors lead to severe developmental diseases (Brown and Brown, 2004; Messaed and Rouleau, 2009), it is also possible that longer expansions (>50) of GCG repeat in the alanine frame could be especially deleterious and thus, not represented in late onset inherited neurological diseases such as OPDM and OPML.'

It is not clear to me what the authors are saying here. Are they suggesting that long polyalanine peptides might be embryonic lethal and therefore not seen? If so, would the authors please simply state this clearly.

Line 691: 'villainous' is too anthropomorphic. Please change this.

Lines 704-706: "it is notable that expansion of these GGC repeats over a threshold limit (~200-300 repeats) induces DNA methylation changes, ultimately resulting in silencing of their promoter."

This statement needs a reference

Lines 710-713: fragile X syndrome and Baratela-Scott syndrome (BSS) require references.

Line 724: 'TMPyP4 has no apparent deleterious effect on global cellular transcription and translation.'

The authors need to state whether TMPyP4 is in use clinically.

Line 1006: 'humain'

Line 1009: is 'sectioned' the right word here?

Line 1110: 'scrapped' should be 'scraped'.

Line 1123: 'scrapped' should be 'scraped'.

Line 1283: 'Institute of Genetics and Molecular and Cellular Biology'
Which city?

Line 1300: remove 'both'.

Figures

Figure 2 legend: please state the expression is in HEK293 cells.

Figure 4 legend: please state what stage of muscle differentiation the LHCN-M2 muscle cells have reached after 4 days of differentiation. Are they myoblasts, myotubes? I doubt they are striated myofibers.

Figure 6B – LOC6polyg-GFP is missing an 'l'.

Supplemental information

Supplemental Figures S4, S5, S6 and S7 are mislabelled as S6, S7, S8 and S9.

In the Supplemental S5 legend panel H is mislabelled as 'A'.

Supplemental Figure S6 C: alone of all the mouse models, the CNS expressed uN2CpolyG-GFP mice appear to be hyperactive at 3 months post injection, which is shortly before they die (Figure 6D). Would the authors please comment on this. Are these mice hyperactive?

Videos

There needs to be somewhere a description, legend, of what the videos are showing. What are the GFP structures that appear and disappear as the cells with the inclusions die. How do the cells die? Is it apoptosis? If it is apoptosis, please say so.

Version 1:

Decision Letter:

Our ref: NG-A68278R

21st August 2025

Dear Nicolas,

Your revised manuscript "GGC trinucleotide repeat expansions hidden within small ORFs of the human "non-coding" genome are translated into toxic polyglycine proteins in oculopharyngodistal myopathy" (NG-A68278R) has been seen by the original referees. As you will see from their comments below, they find that the paper has improved in revision, and therefore we will be happy in principle to publish it in Nature Genetics as an Article pending final revisions to satisfy the remaining requests and to comply with our editorial and formatting guidelines.

We are now performing detailed checks on your paper, and we will send you a checklist detailing our editorial and formatting requirements soon. Please do not upload the final materials or make any revisions until you receive this additional information from us.

Thank you again for your interest in Nature Genetics. Please do not hesitate to contact me if you have any questions.

Sincerely,
Kyle

Kyle Vogan, PhD
Senior Editor
Nature Genetics
<https://orcid.org/0000-0001-9565-9665>

Reviewer #1 (Remarks to the Author):

I read with great interest the revised manuscript by Boivin and collaborators. Overall, the authors' revisions have greatly enhanced the clarity and readability, effectively addressing the numerous points raised. In particular, they have clarified the use of the different plasmid constructs and now clearly indicate which experiments were performed in vitro versus ex vivo. In addition, they have added several experiments (e.g., expression of unstable proteins in non-expanded ORFs, comparison of GGC and GGN plasmids) that substantially strengthen the data. I also found their detailed rebuttal very insightful and would encourage the authors to publish it alongside the manuscript, as it contains valuable discussion points that enrich the paper.

My only remaining concern relates to the Supplementary Figures file. The current layout, with many panels spread across multiple pages, makes it difficult to clearly identify corresponding panels. For instance, it was unclear whether Figures S2T, S2U, and S2V, referenced in response to point 17, correspond to the panels labeled T, U, and V on page 8 of the Supplementary Information file. The authors might consider separating the supplementary figures into more important panels suitable for Extended Data Figures and others for inclusion in the Supplementary Information pdf. However, this is a minor issue that should not impede acceptance of the manuscript.

Reviewer #2 (Remarks to the Author):

The authors have answered my questions and dealt with my comments.
I have only a small number of remaining suggestions and comments.

Discussion.

For the new added text of lines 661-684, would the authors please provide references for the leaky scanning mechanism.

I think it would be worthwhile the authors stating that it is intriguing that to date that no expansion of GGA, GGG or GGT glycine codons have been associated with disease. One asks oneself: "Why not?"

Perhaps at line 725:

Although to date, to our knowledge, no expansions of GGA, GGG or GGT glycine encoding codons have been associated with disease, this work questions whether other polyglycine-containing proteins originating from additional GGC-repeat, or other repeat expansions remain to be discovered.

In Supplementary figure S1, to help the reader, please highlight the start codons of the ORFs in the construct cDNA sequences.

Supplementary Figure S7 is still mislabelled as S9.

Strasbourg, 23 July 2025

To: Prof. Kyle VOGAN,
Senior Editor – Nature Genetics

Dear Editor,

Thank you for considering our work for potential publication. Please find below our responses (in blue for clarity) to Referees remarks. Overall, all questions and comments have been addressed, requested data and controls have been added and the text corrected. Notably, we modified the figures 1, 2 and 3 and the discussion to improve their clarity. Moreover, we show that in absence of repeat expansions, the small ORFs identified in our study are nonetheless expressed, however resulting in very small proteins (<100 aas) quickly degraded and thus, hardly detectable in normal conditions (novel supplemental fig. S2Y and S2Z). We also provide key controls showing that optimized GGN repeats have identical RNA and protein expression, as well as identical toxicity compared to pure GGC repeats. Similarly, we show that the TMPYp4 compound corrects toxicity driven by expression of pure GGC repeats identically compared to its effect on optimized GGN repeats. These important controls, which respond to significant points raised by Referee #1, are now presented in supplemental figures S4C-E and S7B-D. Finally, live imaging with single cell tracking data and apoptosis tests, as suggested by Referee #2, have been added in supplemental fig. S4J, S4K and S4L.

We hope that with addition of these controls and corrections, this work will be appropriate for publication. Also, we would like to thank you and the reviewers for their comments and suggestions, which greatly helped us to improve the quality of our manuscript.

Best regards,
Nicolas,

Dr. Charlet Nicolas,
Group Leader,
Head of the Translational Medicine and Neurogenetic Department,
67404 Strasbourg, France
ncharlet@igbmc.fr
+33 388 653 309

REVIEWER COMMENTS:

REVIEWER #1 (Remarks to the Author):

This manuscript by Manon Boivin and colleagues provides compelling new insights into the pathophysiological mechanisms underlying pathogenic GGC repeat expansions associated with oculopharyngodistal myopathy (OPDM) and oculopharyngeal myopathy with leukodystrophy (OPML). GGC repeat expansions ranging from approximately 50 to 200 repeats in at least six genes (LOC642361, LRP12, GIPC1, NOTCH2NLC, RILPL1, and ABCD3) have been linked to autosomal dominant disorders

spanning a clinical continuum between neuromuscular and neurodegenerative diseases. While the expression of polyGlycine peptides by repeat-associated non-AUG (RAN) translation had previously been demonstrated in Fragile X-associated tremor/ataxia syndrome (FXTAS) and NOTCH2NLC-related neuronal intranuclear inclusion disease (NIID) by the same authors (Boivin et al., Neuron, 2021) and others in the field, this study significantly extends these findings, as follows:

- Mechanism of polyG expression: The authors demonstrate that GGC expansions can lead to polyGlycine peptides through the formation of open reading frames (ORFs) initiated either from canonical ATG start codons or from alternative near-cognate codons. These results challenge the current view of how RAN translation works, which was thought to be independently of AUG initiation, and shows that translation occurs in selective reading frames, rather than across all three possible frames.
- Distinct polyG proteins were detected in patient tissues using antibodies targeting the unique N-terminal regions of these peptides. These proteins localize to p62-positive intranuclear inclusions, which constitute pathological hallmarks of GGC repeat expansion disorders.
- Ectopic expression of polyG polypeptides in both *Drosophila* and mouse models recapitulates key skeletal muscle pathologies observed in patients, including the formation of p62-positive inclusions.
- Therapeutic potential: importantly, the authors identify a pharmacological compound capable of selectively reducing polyG peptide expression, offering a possible therapeutic strategy for GGC expansion-related diseases.

The manuscript is generally well written, with a clear presentation of a sound experimental approach and robust results.

Suggested improvements: I listed below points that the authors could address to enhance readability and clarify certain aspects of the experimental design and results.

1) The title does not accurately reflect the content of the manuscript and may even be somewhat misleading. It suggests that translation into toxic proteins is a general mechanism associated with microsatellite expansions across the “dark” genome. As the authors correctly acknowledge, >60 disorders associated with repeat expansion have been described, with diverse underlying mechanisms. RAN translation—or translation of repeats into toxic peptides—is only one of them, and many of these disorders are instead caused by gene loss-of-function due to the repeat expansion (e.g., Fragile X, EPM1/CSTB, Friedreich ataxia/FXN, SCA27B/FGF14, GLS, XYLT1, AFF2, AFF3). Furthermore, this study provides no generalizable evidence that the mechanism of translation applies to microsatellite expansions beyond the specific GGC repeats examined. I therefore strongly recommend revising the title to more accurately reflect the results of the study. I also suggest avoiding the use of the terms “dark genome” or “dark proteome”, which also overstates the broader relevance of the findings.

Thanks, the title was indeed too general and misleading, it is now modified as follow:

“GGC trinucleotide repeat expansions hidden within small ORFs of the human “non-coding” genome are translated into toxic polyglycine proteins in oculopharyngodistal myopathy”.

2) Lines 35 and 60-61 (Summary): the first sentence of the summary is potentially misleading, as it suggests that microsatellites make up half of the genome. As correctly stated in the introduction, microsatellites constitute only about 3-6% of the genome and the majority of repetitive DNA consists of other elements such as LINEs, SINEs, and alpha-satellite sequences. The authors should clarify that the mechanisms described in the manuscript are specific to a subset of microsatellite repeat expansions and are not generalizable to other repetitive elements.

This sentence was indeed confusing and is now corrected as suggested (lanes 35-36):

“3 to 6% of the human genome is composed of microsatellite sequences, which are short DNA elements composed of 1 to 6 nucleotides motifs repeated in tandem.”

3) Lines 104–105: The terms oculopharyngodistal myopathy (OPDM) and oculopharyngeal myopathy with leukodystrophy (OPML) should be spelled out in full and separately at least once in the Introduction, together with their respective OMIM entries. This information currently appears in the Discussion (lines 602–603) but would be more appropriately placed earlier in the manuscript.

This sentence is now amended as recommended, including the OMIM entries that were missing (lanes 101-105):

“Oculopharyngodistal myopathy (OPDM, OMIM #164310) is a rare adult-onset and slowly progressive neuromuscular disease firstly described in 1977 (Satoyoshi and Kinoshita, 1977), while oculopharyngeal myopathy with leukoencephalopathy (OPML, OMIM #618637) is an autosomal dominant disorder with oculopharyngeal myopathy, diffuse limb weakness and leukoencephalopathy described more recently (Ishiura et al., 2019).”

4) Lines 154-156 (Introduction): the references cited in the Discussion (lines 653–654) regarding near-cognate codons should also appear in the Introduction to support the earlier mention of this concept.

References about translation initiation at near-cognate codons are now included in the introduction (lanes 159-160):

(Kozak, 1989; Peabody, 1989, Ingolia et al., 2011; Lee et al., 2012; review in Kears and wilusz, 2017).

5) Lines 185-196 (Results): this section of the Results repeats information already presented in the Introduction and could be significantly condensed or combined with other paragraphs elsewhere to avoid redundancy and improve the flow of the manuscript.

The first paragraph of the results was indeed a recap of the intro and is now simplified as follow (lanes 189-190):

“Identical expansions of GGC repeats located in diverse sequences annotated as non-coding cause Oculopharyngodistal myopathy with or without leukoencephalopathy (OPDM & OPML), however through an unknown mechanism. “

6) A recurring issue throughout the manuscript is the use of clinical terms such as “OPDM” and “OPML” to refer to specific GGC repeat expansion constructs rather than specifying the number of repeats and underlying gene. Both OPDM and OPML are phenotypes resulting from GGC repeat expansions in different genes. Similar GGC expansions at the same locus can lead to distinct disease phenotypes, (e.g., NIID or OPDM linked to expansions in NOTCH2NLC). To avoid confusion, I strongly recommend that the authors refer to the gene context and repeat numbers when reporting experiments performed with plasmids. “OPDM and OPML GGC repeats” used repeatedly (lines 184, 211, 219, 230, 234, 264...) should be avoided. In Figures 1, 2, and 3, terms like “OPML,” “OPDM2,” and “OPDM4” should be replaced with the specific names of the constructs used, similar to “LOC6polyG”, “uGIPpolyG”, and “asRILpolyG” used in Fig. 2F or Fig. 3A-D. The terms OPDM and OPML could be used when patient-derived material from individuals diagnosed with these specific conditions is used.

This was intended to simplify these figures, but ended in just being more confusing. Thus, the text and the figures 1, 2 and 3 have been modified as suggested, now including clear mention of gene names (GIPC1, RILPL1, etc.), RNA elements and repeat localization (antisense, 5’UTR, etc.), GGC numbers and their natures (pure 50 GGC in figures 1 and 2, optimized 100 GGN in fig. 4 and thereafter), disease subtypes, tissue analyzed, etc..

7) It is unclear whether the constructs used in Fig. 1C-G, Fig. 2B-D, Fig. 3-7 are the same or if (and how) they differ as the names “LOC6polyG”, “uGIPpolyG”, and “asRILpolyG” are only used from Fig. 2F on. A schematic summary of all the constructs used in the study, including their names, their features (e.g., gene context, number of GGC repeats) and whether they contain an ATG or near-cognate start codons naturally occurring in the human genome, would greatly improve clarity. Ideally, Fig. 1A, 1B, 2A and 2F could be combined in a single figure or revised as needed to clarify how the constructs differ from one another and from those used in previous experiments.

This is a relevant point, especially as constructs used in figures 1 & 2 are different from the ones in figure 4 and thereafter. Figures 1 and 2 aim at determining whether and how GGC expansions are translated. Thus, these two figures use pure GGC repeats (~50x) embedded in their respective natural upstream transcript sequences. In contrast, figure 4 and afterward aim to study the characteristic of the novel polyglycine proteins. Thus, these figures use optimized (100x GGN) repeats embedded within their respective ORF sequences, thus with their N- and C-terminal flanking amino acids sequences. Of technical interest we used optimized sequences as it is difficult to maintain plasmids with a pure stretch of 100 GGC repeats, and we often ended with mix of shorter expansions, which complexifies the study and comparison of diverse polyGly proteins in parallel. To clarify this point, the figures 1, 2 and 4 have been modified with their respective construct sequences indicated in the supplementary figures 1D-F and 4A. Moreover, the text corresponding to the figure 1 has been modified as follow (lanes 201-206):

“Of technical interest, ~50 pure GGC repeats with their upstream GIPC1 5’UTR, RILPL1 antisense or LOC642361 sequences were cloned and fused to the GFP sequence deleted of its natural ATG start codon, so that expression of the GFP is now dependent of translation initiation occurring directly inside the GGC repeats or within their upstream GIPC1, RILPL1 or LOC642361 hosting sequences (sequences in supplementary figure S1D to S1F).”.

8) Consider revising Fig. 1C-F to incorporate some of the immunofluorescence data and quantifications for GIPC1 and RILPL1 currently shown in Fig. S1. This would ensure consistency with Fig. 1A and 1G, which present data for all three constructs, whereas Panels B-F currently focus only on LOC642361. If space is a limitation, some data (e.g. Panel D) could be moved to the Extended Data.

Figure 1 is now modified to present translation of GGC repeats embedded in either the GIPC1, RILPL1 or LOC642361 sequences alongside each other, resulting in indeed, a much clearer and comprehensive figure.

9) Results, line 21-212: Is there a specific reason why only polyG (and not polyA or polyR) peptides are produced, aside from reading frame selection and initiation codon context? A brief discussion would be helpful.

This point (also noted by Referee #2) is very exciting, but difficult to respond accurately. Translation of GGC repeats in the polyalanine or arginine frames is indeed not observed, suggesting that these microsatellites are not randomly located in ORFs, at least in the OPDM & OPML late onset neuromuscular diseases. Considering that polyarginine-containing proteins expressed in cell and animal models are highly toxic (cf. the C9ORF72 polyGR and polyPR DPR models of ALS), and that expression of proteins with relatively short (~30) stretches of polyalanine lead to severe developmental disorders, one may hypothesize that long expansions of polyalanine or polyarginine (>50 GGC repeats as observed in OPDM & OPML) may not be compatible with late onset, neuromuscular manifestations. However, this is mere speculation. As this provocative point was raised by both Referees, we tentatively added the following text in the discussion (lanes 706-720):

“we detected no overt translation in the alanine or arginine frames. An explanation would be that our assays are not sensitive enough to detect low level of RAN translation with non-canonical initiation starting directly within the repeats and in the three frames. Alternatively, expression of polyalanine or

polyarginine-containing proteins may be too deleterious to be observed in late-onset diseases such as OPDM and OPML. As a support of this hypothesis, mutations resulting in expression of various transcriptional factors with extended, but relatively short (~30), stretches of polyalanine cause severe developmental diseases (review in Brown and Brown, 2004; Messaed and Rouleau, 2009). Similarly, expression of a PABPN1 protein with an extended run as short as 11 to 18 polyalanine is sufficient to cause the late onset oculopharyngeal muscular dystrophy (OPMD) (Brais et al., 1998; review in Banerjee et al., 2013). These data may suggest that much longer GGC repeat expansions (>50), as the ones observed in OPDM and OPML, would be especially detrimental if translated in the alanine frame, and thus, potentially counter selected."

10) Text in Fig. 2B-D is not readable.

Figure 2 is now modified and hopefully clearer.

11) It is currently difficult to clearly distinguish between experiments conducted using in vitro overexpression of GGC repeats from plasmids and those based on patient-derived material. This distinction is critical, as one of the most important contributions of the study is the detection of polyG proteins in patient tissues, which validates that the observed effects are not merely artifacts of constructs/in vitro experiments. The authors should explicitly state which figures correspond to each experimental context by indicating it in the legend of the figure and in the subtitles included in the results. For example, while lines 301–321 suggest that Fig. 3 presents data from patient muscle tissue, the use of the same construct names (e.g., "OPDM2 (uGIPpolyG)" and "OPDM4 (asRILpolyG)" ...) as those used in previous in vitro experiments and the lack of clarity in the figure legend create confusion.

Figure 3 is now modified to differentiate the genes hosting the GGC repeat expansions from their encoded polyglycine proteins and with now a clear indication of their respective antibody epitopes. We also detailed the tissue and OPDM subtype analyzed by immunofluorescence. Thanks to Referee comments, we hope that this important figure, demonstrating existence of these polyglycine proteins in patient tissues, is now clearer and more comprehensible.

12) Results, lines 342–347: The authors introduce modified constructs containing GGN interruptions, which are intended to stabilize the GGC repeat tracts and reduce their instability. These interruptions likely also alter the RNA secondary structure, potentially impacting RNA toxicity. However, it is unclear in which experiments these interrupted constructs were used. The Results text (lines 342–350) implies their use begins in Figure 4, but since the figure labels and construct names do not specify whether interruptions are present, it remains ambiguous whether comparisons were made with their non-interrupted counterparts.

This is a critical issue, as repeat instability in specific brain tissues has recently been implicated in the pathology of several disorders (e.g., CAG expansions in HTT, PMID: 39824182; AAG expansions in FGF14, PMID: 39378335). Moreover, RNA secondary structures are known to contribute to the disease mechanisms in various repeat expansion disorders such as DM1 and CANVAS. Therefore, it is essential that the authors clarify the use of interrupted constructs and, ideally, compare their impact to that of pure repeats. Alternatively, the authors should provide supporting evidence from patient material demonstrating that GGN-interrupted expansions in OPDM/OPML are pathogenic and associated with comparable clinical phenotypes to those caused by uninterrupted repeats.

This is a crucial point as toxicity of an expressed microsatellite expansion could be driven by its RNA or its encoded protein (or a mix of both). Thus, we clarified what construct was used in what assay (cf. comment 7, novel figures 1 and 4 with their respective sequences in supplemental figures 1D-F and 4A). Also, we added experiments showing that a construct with a pure GGC expansion (100 repeats) present similar RNA and protein expression, as well as identical toxicity compared to the same construct and repeat size but with an optimized GGN expansion. Also, and this is a technical detail, but

different protein partners, and have varying impacts when delivered via rAAV vectors in animal models could be interpreted as supporting the latter hypothesis. It would be valuable if the authors could comment on this point in the Discussion, particularly considering their in vivo results.

Another excellent point, which originates and was highly debated in the polyQ field. Thus, we added live microscopy with single cell tracking analyzes of cell death and polyG inclusions formation (directly inspired from the Finkbeiner's studies on huntingtin polyQ toxicity, Arrasate et al., Nature 2004). Of interest, we noted that, as observed for polyQ, presence of polyG aggregates does not correlate with cell death, with observation of numerous dying cells showing a clear diffuse polyG signal. However, we also noted that cells with inclusions are not particularly protected against cell death. Moreover, and as commented by Referees, these diverse polyG proteins show distinct biological properties, with notable differences in their toxicity and aggregation. Consistent with these disparities, quantification show that cell death is not plainly associated with diffuse or aggregated polyGly, and moreover, varies from one protein to another. These novel data, which underscore the importance of the specific amino acid sequences flanking the core polyglycine of these proteins, are presented as supplemental figure 4J with the following description (lanes 442-454):

“Live cell tracking indicates that cell death was observed both in cells with polyGly inclusions, as well as in cells showing a diffuse localization of these polyglycine proteins (supplementary figure S4J); however, with some notable disparities between these proteins, likely originating from their different abilities to form aggregates. Formation of polyGly inclusions is abrupt, with a diffuse localization observed during dozens of hours and sudden aggregation in minutes. Of curiosity, formation of some polyGly aggregates was even observed after cell death. Finally, no overt signs of apoptosis were noted, suggesting that these proteins are toxic by another pathway (supplementary figures S4K and S4L). Interestingly, these data suggest that polyGly inclusions are neither essential to drive toxicity, nor especially protective, highlighting the need to study further the toxicity and the biochemical properties and dynamic of aggregation of these polyglycine proteins. Moreover, these results also stress the importance of the specific amino acid sequences flanking the central polyglycine core of these proteins to modulate their toxicity.”

According to the importance of this debate in the polyQ field, we also added the following discussion (lanes 759-767):

“Of interest, our data suggest that formation of polyGly inclusions are neither critical to induce cell death, nor especially protective, at least in cell models. This observation contrasts with the polyQ proteins, where inclusions are found to be protective by sequestering and thus, decreasing levels of toxic soluble polyglutamine species (Saudou et al., 1998; Klement et al., 1998; Arrasate et al., 2004). Such discrepancy could be related to the biological disparities between the diverse polyGly proteins studied here, and/or caused by the higher propensity of polyglycine to aggregate over polyglutamine. Nonetheless, it remains to meticulously investigate whether these polyGly proteins are pathogenic in their aggregated form, or under their soluble form.”

14) One of the most exciting aspects of this manuscript is the identification of a compound, TMPyP4, that reduces the aggregation and toxicity of polyG proteins. TMPyP4 is best known as a stabilizer of DNA G-quadruplex (G4) structures. Could the authors clarify whether the engineered interruptions introduced into the GGN repeats involve the insertion of adenines (A), which could potentially promote the formation of G4 structures not naturally present in the uninterrupted GGC repeat expansions found in patients? This point is important to assess the relevance of the TMPyP4 findings to the native pathogenic repeats.

This is indeed a key question, notably to envisage TMPyP4 as a potential therapeutic proof of concept for various diseases caused by similar (but not identical) GC-rich microsatellite expansions. As GGN interruptions, notably GGT or GGA, are common in OPDM and OPML (cf. point 12), we believe that our optimized GGN constructs are acceptable tools to model these diseases. Moreover, we added novel

data showing that TMPyP4 similarly corrects polyglycine toxicity, whether this protein is encoded by a pure GGC repeat expansion or by a GGN-interrupted expansion (both constructs with an identical backbone and repeat size, etc.). These important controls showing identical TMPyP4 effect on pure GGC vs. optimized GGN repeats are shown in novel supplemental figures (S7B, S7C and S7D) and discussed as follow (lanes 610-615):

“As a control, since we employed optimized vectors carrying GGN repeats, which may modify secondary RNA structures and thus binding of the TMPyP4 molecule compared to pure GGC repeats, we confirmed that TMPyP4 treatment reduces toxicity and expression of a polyglycine protein encoded by a pure GGC repeat expansion at similar levels compared to the same construct, but with an optimized GGN repeat expansion (supplementary figures S7B to S7D).”

15) Discussion, lines 606-607: there any evidence that GGC repeats within the normal range are translated, or do these GGC repeats form an ORF only in pathological/expanded conditions?

This is a central point as RAN translation reports translation only when repeats are expanded beyond a threshold limit, while here, microsatellites are embedded in classical ORFs, which should be translated whatever their repeat size. To clarify this point, we added in the supplementary figures 2Y and 2Z data showing that these small ORFs with control repeat size (10 GGC) are indeed translated, but into small and unstable peptides, which are detectable only upon inhibition of the cell degradation pathways (with MG132 and bafilomycin treatment). These results suggest that translation of these small ORFs occurs independently of the length of their GGC repeats. However, in absence of an expansion, these very small proteins (<100 amino acids) are unstable and thus hardly detected. These novel data are discussed as follow (lanes 295-302):

“Finally, we noted that these small ORFs with a control repeat size (~10 GGC) are translated, but into small and unstable peptides, which are hardly detected without inhibition of the cell degradation pathways (supplementary figure S2Y). In contrast, these small ORFs become stable when carrying an expansion of their polyglycine stretch (supplementary figure S2Z). These results suggest that translation of these small ORFs follows a canonical translation mechanism, which is independent of the length of their GGC repeats. However, in absence of a polyglycine expansion, their encoded peptides are of small size (<100 amino acids) and thus, unstable and hardly observable.”

16) As briefly discussed in the discussion, not all GGC expansions result in translation into polyG peptides. Only mid-size expansions (50–200 repeats) support translation, while longer expansions are often methylated, leading instead to gene silencing and loss-of-function. It would be helpful if the authors could clearly indicate this distinction and briefly discuss the relationship between expansion size and its functional consequences i.e., translation and hypermethylation-associated silencing a part of the discussion paragraph appearing lines 705-713.

This is indeed a key point discriminating the OPDM and OPML disorders from other repeat expansion diseases with an anticipation mechanism (the size of the repeat expansion correlates with an increased severity and/or an earlier age of onset). In that aspect, note that hypermethylation associated with promoter silencing in asymptomatic individuals who are carrier of ultra long GGC repeat expansions support a gain of function model in OPDM & OPML. Thus, a tentative discussion was added as follow (lanes 801-815):

“...somatic expansion of CAG repeats is associated with the specific degeneration of striatal neurons in Huntington’s disease (Handsaker et al., 2025), questioning whether a similar mechanism may underline tissue specificity in OPDM and OPML. However, there are limited evidence of anticipation in these diseases, and in contrast various studies are now reporting asymptomatic individuals carrying ultra-long (>300) GGC repeat expansions in OPDM and NIID families. These long GGC expansions are associated with DNA hypermethylation and promoter silencing compared to the shorter, transcribed and pathogenic GGC repeat expansions found in individuals with OPDM and OPML (Deng et al., 2020;

Ogasawara et al., 2020; Yu et al., 2021; Kumutpongpanich et al., 2021; Fukuda et al., 2021). Thus, and in opposition to the anticipation observed in Huntington disease, myotonic dystrophy type 1 and various other microsatellite diseases, expansion of GGC repeats beyond a threshold limit (>300) appear to be associated with a protective epigenetic silencing mechanism in OPDM, OPML and NIID. Of interest, these observations strengthen our model that these GGC repeat expansions are pathogenic through a gain-of-function mechanism with expression of a toxic product, and not through a promoter silencing and loss-of-expression mechanism.”

17) Could the authors provide more detail on the expression profiles of the six genes (LOC642361, LRP12, GIPC1, NOTCH2NLC, RILPL1, and ABCD3) in which GGC repeat expansions are linked to the OPDM phenotype? Are all these genes similarly expressed in both muscle and brain tissues? Do the authors expect the ORFs generated by GGC expansions to follow the expression patterns of their host genes, or could they exhibit distinct regulation? Based on the data presented in this manuscript, could the authors comment on whether differential tissue expression of polyG peptides might underlie phenotypic variability observed across disorders such as OPDM, OPML, NIID, or SCA4 (in the case of GGC expansions in ZFH3).

Origin of the tissue specificity is indeed a most challenging debate in the field of neurological diseases (cf. C9ORF72, SOD1, SMN, etc. which are ubiquitously expressed while their associated mutations affect specific neuronal populations). Pattern of *GIPC1*, *RILPL1* and *LOC642361* RNA expression are now presented in supplemental figures S2T, S2U and S2V, showing that OPDM and OPML (as well as NIID and SCA4) make no exception, with a tissue specificity that cannot be accounted by the sole expression pattern of the genes hosting their mutations. Thus, a tentative discussion was added as follow (lanes 788-798):

“Finally, the relationship between the expression pattern of these GGC repeats, the size of their expansions and their toxicity is currently unclear. Of interest, these mutations are embedded in diverse genes with variable tissue distribution and expression levels. Analysis by RT-qPCR and exploration of RNAseq databases indicate that the GIPC1, RILPL1 antisense and LOC642361 RNAs are expressed in skeletal muscles and the central nervous system, but are not limited to these tissues. Moreover, p62-positive inclusions are widely observed outside the skeletal muscles and the nervous system in individuals with NIID (Yamaguchi et al., 2018; Chen H. et al., 2020). Thus, it remains to investigate why muscle cells and neurons are especially sensitive to these mutations. One may also hypothesize that in absence of cell division and its associated nuclear membrane break, muscle cells and neurons may accumulate more toxic polyglycine proteins in their nuclei, compared to other tissues with dividing cells. ...”

REVIEWER #2 (Remarks to the Author):

This manuscript addresses the fascinating and intriguing genetics of oculopharyngodistal myopathy (OPDM). For some time now it has been known that GGC/CCG repeat expansions in the 5'UTR of six, perhaps seven, different genes cause phenotypically very similar oculopharyngodistal myopathies (OPDM) with or without leukodystrophy (OPML). This has always suggested that the pathomechanism has to do with the GGC/CCG repeat expansions and the genes that the expansions are in the 5'UTR of are, in fact, irrelevant. This manuscript demonstrates that the pathomechanism is transcription and translation of polyglycine-containing proteins from short previously unidentified open reading frames in the 5'UTRs.

The authors have done an extraordinary amount of work:

- They focussed their experimental work primarily on three of the OPDM genes: GIPC1, RILPL1 and LOC642361.

- They expressed the GGC repeat expansions of GIPC1, RILPL1 and LOC642361 and control size repeats in their genomic context fused to GFP in HEK293 cells in their Glycine, Alanine and Arginine frames. Only the polyglycine frame resulted in protein detectable with GFP antibodies despite RT-qPCR demonstrating similar RNA levels.
- They used mass spectrometry to sequence the polyglycine proteins and showed the N-terminal sequences are different, reflecting the sequence of the gene, but all start from a standard acetylated methionine.
- Using antibodies to the specific N or C-terminals they demonstrated the presence of the polyglycine proteins for GIPC1, RILPL1 and LOC642361 but also OPDM3 NOTCH2NLC in the p62 inclusions in muscle biopsies from the respective patients.
- To test for RAN translation in other potential frames, they developed an antibody specific to the GIPC1 alanine frame but did not see any staining in OPDM2 biopsies.
- They expressed the polyglycine proteins in human LHCN-M2 differentiated muscle cells and showed they formed p62 positive cytoplasmic and intranuclear inclusions. There were some differences in the results in that some of the constructs resulted in more cytoplasmic or nuclear inclusions than others.
- Immunoprecipitation and mass spectrometry identified interactants that varied between the genes, suggesting the unique N- or C-terminals were involved in the interactions.
- Testing the toxicity of the polyglycine proteins in LHCN-M2 differentiated muscle cells showed all were toxic but the RILPL1, NOTCH2NLC and LOC642361 more than GIPC1.
- To test pathogenicity in animal models, they expressed the polyglycine proteins in C57BL/6 mice using a myotropic virus MyoAAV 4A which can be used to reach muscle after intravenous injection. This resulted in pathology in both tibialis anterior and gastrocnemius including p62 inclusions. Again, there was variability in cytoplasmic vs intranuclear location between the polyglycine proteins. snRNA sequencing did not show major changes. The asRILpolyG (OPDM4) and LOC6polyG (OPML) polyglycine proteins resulted in early death which was shown to be associated with remarkable dilated cardiomyopathy not shown with the other proteins.
- Similarly, to test toxicity in the brain, the authors used an AAV vector PHP.eB rAAV serotype that crosses the mouse blood-brain barrier and targets neurons. This resulted in variably altered performance, loss of neurons including Purkinje cells and early death with the different polyglycine proteins.
- In relation to candidate therapies, the authors tested a number of compounds that have been shown to prevent nuclear export of translation of toxic proteins or promote autophagy of aggregation prone proteins. The authors showed the porphyrin TMPyP4 to be a potential therapeutic option for these polyglycine diseases since TMPyP4 reduced toxicity in tissue
- Finally in animal models the authors expressed the polyglycine proteins in Drosophila. This resulted in eye, locomotor phenotypes and premature death.

Major comments

1) To my mind, the major thing missing from the manuscript is a detailed discussion of the pathogenetic mechanism of the diseases. I was expecting in the Discussion a detailed description of what the manuscript demonstrates are the prerequisites for the oculopharyngodistal myopathy phenotype and this detailed explanation is simply not there. The authors demonstrate that the prerequisites are:

- There has to be a transcript
- There has to be a cognate or near-cognate start codon with an associated Kozak sequence
- The start codon has to be in the frame with the GGC repeat that can then be translated into polyglycine.

This is indeed a key point that was absent from our manuscript, especially considering the RAN translation mechanism (translation initiating directly within the repeat expansion and in the three possible frames). Discussion is now modified as follow (lanes 662-669):

“In addition, this work clarifies some prerequisites for these GGC microsatellite expansions to be translated in a stable and detectable polypeptide, notably the necessity for these repeats to be (i) embedded in a RNA transcript, which may include ill-described sequences annotated by default as non-coding, but with the requirement that this RNA is exported within the cytoplasm where translation occurs ; (ii) embedded within an ORF, with the crucial point to be in frame with an upstream cognate ATG or near-cognate (ACG, CTG, GTG or TTG) start codon with its associated Kozak motif and (iii) of sufficient size for the encoded polypeptide to be stable and thus, reliably detected in patient materials.”

2) The authors beautifully demonstrate this in that for the GIPC1 and LOC642361 the (GGC)_n repeat, is in the sense strand, whereas for RILPL1 the sense strand repeat is (CCG)_n and is not translated into polyproline, polyalanine or polyarginine, but there is bidirectional transcription of RILPL1 and it is the antisense transcript encoding a (GGC)_n repeat that gives rise to the polyglycine pathogenic protein.

Their needs to be discussion of why this pathogenetic mechanism occurs in 5'UTR but to date has not been identified in 3'UTR.

This is another interesting point, related to the ribosome scanning mechanism from the 5' toward the 3' direction with dissociation of ribosomal subunits at stop codons (with no, or rare and tightly regulated, reinitiation in mammals). These key features of the translation machinery explain the rarity of translated polypeptides originating from 3'UTR sequences, and highlight why most translated microsatellites expansions are found in 5'UTR sequences (cf. this work in OPDM2 and OPDM3), or within classical ORFs (cf. GGC repeats in the ZFH3 ORF in SCA4 and, of course, the polyQ proteins). A short overview of these key features of the translation machinery is now discussed as follow (lanes 670-683):

“An additional point is the localization (5' vs. 3'UTR) of these potential ORFs, which is dictated by the 5' to 3' ribosome leaky scanning mechanism. Briefly, when encountering a start codon located in an unfavorable context (poor Kozak sequence, near-cognate start codons, etc.) a ribosome may either initiate translation, or bypass this codon, resumes its scanning and begins translation at another downstream start codon. Of interest, selection of these alternative start codons can be regulated by regulatory proteins, specific RNA motifs and structures, notably secondary RNA structures shaped by repeat expansions. This leaky ribosomal scanning mechanism multiplies and complexifies the number of ORFs encoded by our genome, but is also a major source of regulation, most often downregulation in case of upstream ORFs. This is prompted by the translation termination mechanism, where dissociation of the ribosomal subunits at an upstream ORF stop codon generally results in the loss of this ribosome for translation of downstream ORFs. These key features illuminate why most translated polypeptides and small ORFs are found in 5'UTR sequences, especially compared to 3'UTR sequences, and overall, support translation of the GIPC1, NOTCH2NLC, RILPL1 and LOC642361 GGC repeat expansions in one predominant frame.”

4) In the Discussion, at present there is a very brief discussion (Lines 638-670) that, now that many small ORFs and microsatellites have been identified in the genome, some might be similarly pathogenic to those causing OPMD/OPML. This might be written more succinctly and needs to be offset by the discussion of the stringent multiple requirements for toxic proteins to be produced.

Indeed, our comment required to be tempered as it may give the false impression that any repeat expansion would be translated and potentially pathogenic, which is erroneous considering other mechanisms of diseases, notably promoter silencing and loss of expression of the gene hosting these repeats and /or RNA gain of function. The discussion has thus, been modified as follow (lanes 739-742):

“Thus, as the human genome contains ~2 millions of microsatellites, which populate up to 6% of our DNA, it is thus foreseeable that some microsatellites will inevitably fall in one of these numerous, small and ill-described ORFs. Provided that these repeat expansions are embedded in transcribed and exported RNAs, as well as located in frame with an upstream start codon in a correct Kozak environment, our present observation of novel polyglycine proteins expressed from GGC microsatellite expansions may be only the tip of the iceberg.”

5) There is no discussion of the fact that multiple other codons in the sense strand could make polyglycine if expanded into a repeat: ie. CGG (Arg), GAG (Glu), GCG (Ala), GGA, GGG and GGT (Gly), GTG (Val), or that there are multiple codons if expressed from the reverse strand and expanded into a repeat, could be translated into polyglycine. There should be some discussion as to why this has not been seen.

6) There should be greater discussion of what the mechanism might be that, despite RT-PCR showing equal expression of the expression vectors, only the polyglycine frame resulted in protein being detectable. What is happening to the polyalanine and polyarginine proteins?

These two points, also raised by Referee #1, are particularly appealing, especially considering the RAN translation mechanism (initiation directly within the repeats and in the 3 possible frames). While it is difficult to discuss other repeat expansions that we have not investigated here, such as RAN translation of CTG/CCTG repeats reported in myotonic dystrophy or of CAG repeats in Huntington’s disease, etc., we can tentatively discuss GGN repeats translation:

- First, there is no (to our knowledge) reports of diseases caused by expansions of microsatellite composed of GGA, GGT or GGG repeats, but indeed one may hypothesize that such expansions, if embedded in a transcribed RNA and in frame with a start codon, would potentially be translated in polyglycine.

- A second intriguing point are some Fragile sites associated with GGC repeat expansions located in promoter and/or 5’UTR sequences and that cause promoter silencing and loss of expression of the gene hosting these repeats (cf. GGC expansions in the ZNF713, C11ORF80, DIB2 and XYLT1 genes causing respectively FRA7A, FRA11A, FRA12A and BSS, etc.). One may hypothesize that some of these GGC expansions of intermediate size may not induce promoter silencing, and thus, would be expressed at the RNA level (and with the additional requirement to be in frame with a start codon; cf. Referee #2 first point) could be translated in putative toxic proteins. This hypothesis is inspired by the Fragile X Tremor Ataxia Syndrome (FXTAS) where intermediate size of GGC repeats (50 to 200) are transcribed and translated in a toxic polyglycine protein, while larger expansions (>200 repeats) induce DNA epigenetic changes, promoter silencing and thus, loss-of-expression of the FMR1 encoded protein (FMRP) in the Fragile X Syndrome. This is an exciting hypothesis, however, beyond FXTAS, we are not aware of neurological diseases associated with intermediate size of GGC repeat expansions in the ZNF713, C11ORF80, DIB2, XYLT1, etc. genes.

- Third, we do not observe translation of the *GIPC1*, *NOTCH2NLC*, *RILPL1* and *LOC642361* GGC repeats in the polyalanine or arginine frames, suggesting that these microsatellites are not randomly placed in ORFs, at least in OPDM and OPML. It is always difficult to discuss a lack of observation, but beyond a limitation of our experimental assays, one may also hypothesize that these frames would be either neutral and not toxic; or conversely, too deleterious with expansions over 50 repeats and thus, counter selected and/or may lead to more severe and/or earlier onset clinical manifestations (cf. the early onset and severe SPD1, BCCD, HFGS, BPES, EIEE1, CCHS, etc. syndromes caused by stretch of only ~30 polyalanines in the HOX13D, RUNX, HOXA13, FOXL2, ZIC2, ARX, PHOX2B, etc.,).

- Last but not least, one may also hypothesize that translation may shift from one frame to another, resulting in chimeric polypeptides with different biological properties and toxicity. However, we found no evidence of ribosomal frameshifting during translation of GGC repeats in our models. Moreover, long read sequencing show no overt indels (+1 or -1 nucleotide) within GGC repeat expansions in

individuals with OPDM and OPML. Thus, a shift from the glycine to the alanine or arginine frame, resulting in expression of potentially toxic polyGlyAla (polyGA) or polyGlyArg (polyGR) chimeric proteins is unlikely. Nonetheless, this remains an exciting hypothesis for other repeat expansions (cf. recent long read sequencing in C9ORF72 showing multiple -1 indels in the GGGGCC repeats, which thus may shift their translation, Udine et al., 2024).

Overall, these various points are now also discussed as follow (cf. also point 9 of Referee #1, manuscript lanes 706-722):

“...we detected no overt translation in the alanine or arginine frames. An explanation would be that our assays are not sensitive enough to detect low level of RAN translation with non-canonical initiation starting directly within the repeats and in the three frames. Alternatively, expression of polyalanine or polyarginine-containing proteins may be too deleterious to be observed in late-onset diseases such as OPDM and OPML. As a support of this hypothesis, mutations resulting in expression of various transcriptional factors with extended, but relatively short (~30), stretches of polyalanine cause severe developmental diseases (review in Brown and Brown, 2004; Messaed and Rouleau, 2009). Similarly, expression of a PABPN1 protein with an extended run as short as 11 to 18 polyalanine is sufficient to cause the late onset oculopharyngeal muscular dystrophy (OPMD) (Brais et al., 1998; review in Banerjee et al., 2013). These data may suggest that much longer GGC repeat expansions (>50), as the ones observed in OPDM and OPML, would be detrimental if translated in the alanine frame, and thus, potentially counter selected. Moreover, we found no overt evidence of ribosomal frameshifting during GGC repeats translation, neither we found +1 or -1 nucleotide indels in long read sequencing of DNA from OPDM and OPML individuals, suggesting that a frameshifting from the glycine to another frame resulting in expression of putative polyGly-Ala or polyGly-Arg chimeric proteins is unlikely.”

Minor comments

- Line 138: Nobody uses “similitudes”, similarities would be plainer English.
- Line 213: change spelling to “negligible”
- Line 213: For Figure 1C and all other figures involving microscopy, please add the magnifications used.
- Lines 235-237: “To uncover how these GGC repeats are translated, we immunoprecipitated the OPDM and OPML polyglycine proteins and determined their N-terminal sequences by mass spectrometry analysis.” Please state here in the text that it is from the expression in HEK cells to make it clear to the reader, so the reader doesn't have to search in the Figure legends to find this out.
- Line 558: remove ‘both’.
- Lines 558-559: ‘TMPyP4, which efficiently prevents expression of both uGIPpolyG, uN2CpolyG, asRILpolyG and LOC6polyG at the protein level”

Does TMPyP4 prevent expression, or translation? Would it not be better to write that TMPPyP4 reduces abundance of the polyG peptides? This would also be more consistent with the statement later in the paragraph that TMPyP4 acts principally on translation.

- Line 606: ABCD3 needs to be added as in the Introduction.
- Lines 629-633: ‘Short (~30) stretches of polyalanine in various transcription factors lead to severe developmental diseases (Brown and Brown, 2004; Messaed and Rouleau, 2009), it is also possible that longer expansions (>50) of GCG repeat in the alanine frame could be especially deleterious and thus, not represented in late onset inherited neurological diseases such as OPDM and OPML.”

It is not clear to me what the authors are saying here. Are they suggesting that long polyalanine peptides might be embryonic lethal and therefore not seen? If so, would the authors please simply state this clearly.

- Line 691: ‘villainous’ is too anthropomorphic. Please change this.

- Lines 704-706: "it is notable that expansion of these GGC repeats over a threshold limit (~200-300 repeats) induces DNA methylation changes, ultimately resulting in silencing of their promoter."

This statement needs a reference

- Lines 710-713: fragile X syndrome and Barata-Scott syndrome (BSS) require references.

- Line 724: 'TMPyP4 has no apparent deleterious effect on global cellular transcription and translation.'

The authors need to state whether TMPyP4 is in use clinically.

- Line 1006: 'humain'

- Line 1009: is 'sectioned' the right word here?

- Line 1110: 'scrapped' should be 'scraped'.

- Line 1123: 'scrapped' should be 'scraped'.

- Line 1283: 'Institute of Genetics and Molecular and Cellular Biology'

Which city?

- Line 1300: remove 'both'.

Figures

- Figure 2 legend: please state the expression is in HEK293 cells.

- Figure 4 legend: please state what stage of muscle differentiation the LHCN-M2 muscle cells have reached after 4 days of differentiation. Are they myoblasts, myotubes? I doubt they are striated myofibers.

- Figure 6B – LOC6polyG-GFP is missing an 'l'.

Supplemental information

- Supplemental Figures S4, S5, S6 and S7 are mislabelled as S6, S7, S8 and S9.

- In the Supplemental S5 legend panel H is mislabelled as 'A'.

- Supplemental Figure S6 C: alone of all the mouse models, the CNS expressed uN2CpolyG-GFP mice appear to be hyperactive at 3 months post injection, which is shortly before they die (Figure 6D). Would the authors please comment on this. Are these mice hyperactive?

Videos

- There needs to be somewhere a description, legend, of what the videos are showing. What are the GFP structures that appear and disappear as the cells with the inclusions die. How do the cells die? Is it apoptosis? If it is apoptosis, please say so.

1 - Thanks for these comments and helpful corrections. All typo and grammar have been corrected.

2 - Concerning uN2CpolyG-expressing mice, this is a neat comment and indeed, these animals present some intriguing differences compared to other polyG-expressing mice, and notably signs of hyperactivity. These data stress that despite having a common polyglycine core, these different proteins show diverse pathogenicity. Thus, the text has been amended as follow (lanes 545-548):

"Of interest, mice expressing the uN2CpolyG protein travel an increased distance in open field at 3 months post-AAV injection, suggesting some hyperactivity (supplementary figure 6C). This behavior was not observed in other polyG-expressing animals, stressing that despite a common polyglycine core, these diverse proteins diverge in their pathogenicity."

3 - Concerning videos, their description was indeed unclear and vague. Briefly, they show U2OS cells expressing the NUP50-Cherry protein to stain cell nuclei and the GFP-tagged asRILpolyglycine (OPDM4) protein. Of interest, these images indicate that polyG inclusions are not imported within the nucleus (at least under the form of visible aggregates), but may form directly within cell nuclei and expand. Thus, the video legend was clarified, and the text was modified as follow (lanes 386-388):

“Of interest, live imaging suggests that large polyglycine inclusions don’t migrate toward and penetrate nuclei, but may directly aggregate and grow within the nucleus (supplementary videos 1 and 2).”

4 - Next, we extended these live imaging with single cell tracking and noted that presence of polyGly inclusions was not obligatory associated with cell death, but also that these inclusions were not overtly protective. This contrasts with polyQ proteins that are reported as more toxic under their soluble non-aggregated forms (cf. also point 13 of Referee #1). However, this is a mere correlation and could be linked to the different properties of the diverse polyglycine proteins studied here (notably their difference in aggregation), and/or linked to their propensity to aggregate more easily and faster than polyglutamine. These novel data are now presented in supplementary fig. S4J with the following discussion (lanes 758-767):

“Of interest, our data suggest that formation of polyGly inclusions are neither critical to induce cell death, nor especially protective, at least in our cell models. This observation contrasts with the polyQ proteins, where inclusions are found to be protective by sequestering and thus, decreasing levels of toxic soluble polyglutamine species (Saudou et al., 1998; Klement et al., 1998; Arrasate et al., 2004). Such discrepancy could be related to the biological disparities between the diverse polyGly proteins studied here, and/or caused by the higher propensity of polyglycine to aggregate over polyglutamine. Nonetheless, it remains to meticulously investigate whether these polyGly proteins are pathogenic in their aggregated form, or under their soluble form.”

5 - Finally, these live imaging show that at later time points the nucleus morphology changes, with evident condensation, fragmentation and nuclear membrane rupture, which are indicative of cell death. However, these alterations are not specific to a peculiar cell death mechanism and are observed during both apoptosis and necrosis. Thus, we investigated classical markers of apoptosis (annexin 5 and cleaved caspase 3), but observed no overt signs of apoptosis, suggesting that polyG proteins are driving cell death through a different pathway (most likely necrosis, but this remains to be thoroughly tested). These novel data are now presented as supplemental figures 4K and 4L, and the text has been modified as follow (lanes 447-448):

“Finally, no overt signs of apoptosis were noted, suggesting that these proteins are toxic by another pathway (supplementary figures S4K and S4L).”

Strasbourg, 11 November 2025

To: Prof. Kyle VOGAN,
Senior Editor – Nature Genetics

Dear Editor,

Thank you for considering our work for publication. Please find below our responses (in blue for clarity) to the final Referees comments. Moreover, we have edited our manuscripts and figures according to *Nature Genetics* guidelines and recommendation (text <4400 words, 8 figures with 7 extended data figures, supplemental information in one single PDF file, etc.). We hope that this work is now appropriate for publication. Also, we would like to thank you and the reviewers for their comments and suggestions, which greatly helped us to improve the quality of our manuscript.

Best regards,
Nicolas,

Dr. Charlet Nicolas,
Group Leader,
Head of the Translational Medicine and Neurogenetic Department,
67404 Strasbourg, France
ncharlet@igbmc.fr
+33 388 653 309

REVIEWER COMMENTS:

REVIEWER #1 (Remarks to the Author):

I read with great interest the revised manuscript by Boivin and collaborators. Overall, the authors' revisions have greatly enhanced the clarity and readability, effectively addressing the numerous points raised. In particular, they have clarified the use of the different plasmid constructs and now clearly indicate which experiments were performed in vitro versus ex vivo. In addition, they have added several experiments (e.g., expression of unstable proteins in non-expanded ORFs, comparison of GGC and GGN plasmids) that substantially strengthen the data.

I also found their detailed rebuttal very insightful and would encourage the authors to publish it alongside the manuscript, as it contains valuable discussion points that enrich the paper.

My only remaining concern relates to the Supplementary Figures file. The current layout, with many panels spread across multiple pages, makes it difficult to clearly identify corresponding panels. For instance, it was unclear whether Figures S2T, S2U, and S2V, referenced in response to point 17, correspond to the panels labeled T, U, and V on page 8 of the Supplementary Information file. The authors might consider separating the supplementary figures into more important panels suitable for Extended Data Figures and others for inclusion in the Supplementary Information pdf. However, this is a minor issue that should not impede acceptance of the manuscript.

We deeply thank the Referee for his comments and help. As also suggested by the Editorial office of Nature Genetics, we modified and reduced the number of panels in the main figures, resulting in the creation of 7 novel Extended Data Figures. Additional data are presented in one supplementary information file. We hope that these changes make our text and figures easier to read and more intelligible. Finally, as suggested by the Referee, we fully agree to have the reviewer comments, author rebuttal letters, and editorial decision letters published as a Supplementary item.

REVIEWER #2 (Remarks to the Author):

The authors have answered my questions and dealt with my comments. I have only a small number of remaining suggestions and comments.

- For the new added text of lines 661-684, would the authors please provide references for the leaky scanning mechanism.

- I think it would be worthwhile the authors stating that it is intriguing that to date that no expansion of GGA, GGG or GGT glycine codons have been associated with disease. One asks oneself: "Why not?" Perhaps at line 725:

Although to date, to our knowledge, no expansions of GGA, GGG or GGT glycine encoding codons have been associated with disease, this work questions whether other polyglycine-containing proteins originating from additional GGC-repeat, or other repeat expansions remain to be discovered.

- In Supplementary figure S1, to help the reader, please highlight the start codons of the ORFs in the construct cDNA sequences.

- Supplementary Figure S7 is still mislabeled as S9.

Many thanks for these helpful comments and observations. References about leaky ribosome scanning have been added (*Marilyn Kozak, 2002; Andreev et al., 2022; Dever et al. 2023*), the discussion about other repeat expansions (GGA, GGT and GGG) has been modified (however, this is now included as a supplementary Note in the Supplementary information file due to editorial size constraint) and the supplementary figures S1 and S7 have been corrected as suggested.

Overall, we would like to sincerely thanks and acknowledge both Referees for their precious suggestions and advice, which undoubtedly helped us to strengthen and ameliorate the quality and intelligibility of this manuscript.